# Study on the Application of Foamed Lightweight Soil in Road Widening Project: A Numerical Insight

**DOI:** 10.3390/ma17225432

**Published:** 2024-11-07

**Authors:** Pu-Hao Li, Ke-Zhen Yan

**Affiliations:** College of Civil Engineering, Hunan University, Changsha 410082, China; yankz@hnu.edu.cn

**Keywords:** mechanically stabilized earth wall, centrifuge model test, foamed lightweight soil, road widening project, harmonic search algorithm

## Abstract

This paper introduces a novel retaining wall structure that integrates a traditional mechanically stabilized earth (MSE) retaining wall with foamed lightweight soil (FLS) as the fill material. To evaluate the performance of the structure, a numerical approach based on the finite difference method was employed. Firstly, numerical models were developed based on a centrifuge test model designed by previous researchers, and the results were compared with the measured data. The close agreement between the experimental values and simulations demonstrates the reliability and validity of the proposed numerical models. Subsequently, a series of parametric studies were conducted to reveal the effect of key parameters on the performance of the newly proposed retaining wall. Furthermore, this paper proposes a modified harmonic search algorithm (MHSA), which is based on the original harmonic search algorithm (OHSA), to optimize the design of the proposed retaining wall structure. The results indicate that the proposed retaining wall structure can effectively reduce the differential settlement between the existing road and the newly constructed road at a relatively lower cost. The MHSA can serve as a practical design guidebook for engineers and potential users, enabling rapid and efficient design.

## 1. Introduction

China is one of the most rapidly developing countries in the world in terms of pavement engineering, and by the end of 2022, the total mileage of roads in China had reached 5.35 million kilometers, of which the total mileage of completed highways had reached 177,000 km, ranking first in the world [1]. To meet the growing regional transport demand and promote modern economic development, it is urgent to optimize and upgrade many existing roads and carry out road-expanding work to improve road operation capacity [2].

There are several treatments for widening the old embankment, such as the mechanically stabilized earth (MSE) retaining wall widening method [3], and pile-support geosynthetic-reinforced embankment. Ren et al. [4] have been using shored mechanically stabilized earth (SMSE) walls in embankment-widening projects due to their excellent mechanical performance, straightforward building process, cost-effectiveness, and minimal site requirements. The advantages of conventional widening treatment are fast construction and low impact on existing traffic at a relatively lower cost. Over the past several decades, however, unacceptable vertical settlement of widening embankments and maximum differential settlement between the old and new embankments have frequently been observed in engineering practice [5,6]. Improper utilization of MSE retaining walls may also cause overall instability of the existing embankment, leading to the failure of MSE retaining walls. Alimohammadi and Memon [7] and Golakiya and Lad [8] pointed out that the MSE retaining wall structure is supposed to be built with a rigid foundation, otherwise, significant differential settlement of road surface may be induced by the deformation of the foundation. To overcome the limitations of traditional treatment (i.e., MSE retaining wall), the pile-supported geosynthetic-reinforced embankment is designed and implemented in practical engineering applications [9,10,11,12,13,14,15,16]. Nunez et al. [17] implemented pile-bearing reinforced embankments on soft soil roadbeds, resulting in a reduction in differential settlement on the road surface of newly constructed roadways. Zhao et al. [18] conducted a case study on the expansion of a highway embankment. They utilized prestressed tubular concrete (PTC) piles and geogrids to strengthen the embankment and decrease the duration of construction. Halder and Singh [19] demonstrated that utilizing geosynthetic reinforcement and pile-supported embankments is a highly effective approach for addressing issues related to total and differential settlement, subsoil pressure, and lateral movement of foundation soils when constructing highway embankments on soft soils with low shear strength. This treatment deals with shortcomings in settlement when using the traditional widening way. Nevertheless, the cost of pile-support geosynthetic-reinforced soil walls is relatively high and cannot be accepted on some low-level roadways. Moreover, the traffic will certainly be affected by driving piles when constructing a new embankment. Hence, a new widening way which combines the advantages of these two treatments is needed.

After reviewing the existing widening ways, it would be an ideal treatment way that combines the advantages of traditional MSE retaining walls and controlling the vertical settlement at a lower cost. Based on this concept, the authors proposed a combined widening structure that uses foamed lightweight soil (FLS) as a fill material for traditional MSE retaining walls. The primary benefits of FLS include high strength at low density, adjustable capacity, pumpability through pipes, self-supporting properties, simplicity of construction, and excellent fluidity [20,21,22,23,24,25,26,27,28,29]. Shi et al. [30] proposed the utilization of lightweight foam concrete (LWFC) as a filler for the extended embankment to reduce the settlement discrepancy between the existing embankment and the widened embankment by reducing its self-weight. Yu et al. [3] also observed the benefits of using lightweight fill to decrease settling in existing embankments. Lu et al. [27] discovered that the use of unique artificial soils, specifically FLS, can effectively minimize differential settling of the subgrade in road construction projects due to their lightweight and strong properties. By properly integrating MSE and FLS retaining walls in the design and implementation of road widening projects, MSE–FLS retaining walls may be developed, thereby leveraging the advantages of each while mitigating their deficiencies. Replacing a portion of the FLS material in the new embankment with a portion of MSE retaining wall can significantly decrease the quantity of FLS required, thereby reducing project costs. Additionally, this approach leverages the properties of FLS to effectively minimize the maximum differential settlement between the existing and new embankment, while simultaneously maximizing the rapid construction capabilities of the MSE retaining wall. Meanwhile, it can utilize the rapid building features of the MSE retaining wall, hence reducing the construction duration. However, to the best of the authors’ knowledge, none of the existing research has been reported before on the MSE–FLS structure, a preliminary investigation on this structure is worth studying.

In addition, it would be useful and meaningful to optimize the structure by employing an algorithm and offer a design guidebook after proposing a new retaining wall structure. It is generally accepted that the cost of retaining walls would be reduced after using an advanced algorithm. For example, Basudhar et al. [31] were the first to use the engineering cost of MSE retaining walls as an objective function and for optimization. The results of the study showed that the optimization of the MSE retaining wall using the sequential unconstrained minimization technique (SUMT) reduced its cost by 7–8% as compared to the conventionally designed MSE retaining wall. Manahiloh et al. [32] used a metaheuristic algorithm of harmonic optimization to optimize the cost of an MSE retaining wall, which showed that the construction cost of an MSE retaining wall optimized using the harmonic optimization technique was reduced by 9.2% compared to SUMT. It can be found that HSA is a well-established algorithm that can provide a design proposal of a retaining wall with a relatively lower cost. However, conventional HSA that assumes the fill material in an expanded embankment is uniform material, which is not suitable to be applied to the newly proposed MSE–FLS structure (the filled material is the combination of FLS layer and soil layer). The explanation can be found in Section 4 on the calculation of stability coefficients. Therefore, a modified HSA is necessary to be developed to optimize the proposed structure and give a quick guidebook for engineers.

Based on the above, this paper proposed a new retaining wall structure that combined the advantages of traditional MSE and newly popular materials in engineering projects (i.e., FLS). Firstly, a numerical model was built based on a well-established centrifuge experiment, followed by verifying the model by comparing the calculation results and experimental data. Secondly, the performance of the proposed retaining wall was examined through parametric studies on some key design parameters, namely the width of the foundation, the buried depth of the wall, the filled place of FLS, and the properties of FLS. Thirdly, a modified HSA was proposed to give a quick reference for engineers and potential users.

## 2. Model Establishment and Verification

To examine the performance of the proposed MSE–FLS structure, this section is going to establish a three-dimensional numerical model with a width of 1 m and verify the correctness and reliability of the model with a centrifuge test.

Some of the following assumptions are made in this model:(1)The embankments and foundation layers are considered homogenous and continuous, adhering to the Drucker–Prager model, with continuous contact between the layers;(2)The foundation surface serves as the permeable boundary, while the remaining boundaries are impermeable;(3)Simultaneous construction occurs on both sides during the widening process.

Figure 1 shows a schematic diagram of centrifuge test using a pile-supported reinforced embankment to widen the existing road. The similarity ratio *n* of the centrifuge model test is 40. The relationship between the similarity ratio and the physical quantities is shown in Table 1.

According to the similarity ratio rule presented in Table 1, a numerical model was developed based on the centrifuge model test depicted in Figure 1. The embankment has a height of 6 m and a width of 24 m, with an extension of 10 m on each side of the road. The foundation has a width of 40 m and a depth of 19 m, consisting of two layers: a 15 m deep soft soil foundation and a 4 m deep sand layer. The slope of the original embankment’s shoulder is 1:1.5, which is reduced to 1:0.5 during the widening process. A 0.5 m thick layer of sand bedding is placed on the soft clay layer, with a geogrid embedded in the middle and fixed at the end of the filled soil. The pile length is 15 m, the pile spacing is 2.5 m, and the pile cap size is 1 m by 1 m by 0.6 m. Due to the structural symmetry, only half of the road’s width, as shown in Figure 2B, is considered in the subsequent numerical analysis. The Drucker–Prager model is employed as the constitutive model for soil and sand [33]. The foundation soil and old embankment reach a state of equilibrium under self-weight. Before the construction of the new embankment, the displacement of the foundation soil and old embankment is reset to zero, while the stresses are maintained and kept unchanged. After constructing the new embankment and pile, a new stress equilibrium is calculated in FLAC until the maximum unbalanced forces satisfy the default setting requirement. Additional soil parameters are summarized in Table 2, while the material parameters of piles and geogrids are presented in Table 3. The parameters selected in both Table 2 and Table 3 are a summary of the parameters from the previous research of Li [33].

Figure 3 depicts the comparison between the test data and calculation results of three monitoring points shown in Figure 2B from the developed numerical model. The vertical settlement of the road surface exhibits a “V” shape, with the maximum settlement occurring at the junction of the old and new embankments. Furthermore, the figure demonstrates a strong agreement between the test data and the calculated results, thereby validating the accuracy and reliability of the numerical model proposed in this study. However, a minor offset will exist between monitoring point 1 and monitoring point 2 compared to the centrifuge test data, and the reason for this offset is caused by the fact that the strength of the foundation piles exceeds the calibrated strength, resulting in reduced settlement data from the centrifuge model at same stress levels, hence causing the numerical simulation of settlement to be greater.

## 3. Parametric Analysis

To further investigate the influence of thickness, density, and the placing position of FLS and the modulus of the foundation as well as the spacing of geogrid on the performance of MSE–FLS retaining wall, this section will undertake a comprehensive parametric analysis to quantify these effects because these parameters may greatly influence the properties of the MSE–FLS retaining wall. The parameters used in this section are listed in Table 4, where the FLS parameters are derived from technical specification for cast in situ foamed lightweight soil: (CECS 249-2008) [34], and the values for the new roadbed fill and leveling pad parameters are both sourced from [33]. Unless otherwise specified, these parameters will remain constant throughout the analysis.

### 3.1. Effect of Setting FLS

Figure 4 illustrates the methodology for roadbed expansion through the MSE widening approach, eliminating the need for piling. Figure 4A demonstrates the widening accomplished solely by the application of an MSE retaining wall, designated as Case 1. Figure 4B illustrates the widening achieved through the integration of MSE and FLS, with FLS functioning as the upper roadbed fill, referred to as Case 2. Figure 4C illustrates the widening achieved by integrating MSE and FLS, with FLS serving as the bedding layer, referred to as Case 3.

Figure 5 displays the surface settlement curves for the four distinct widening procedures mentioned earlier. Based on the figure, it is evident that the use of the MSE retaining wall widening approach significantly decreases the surface settlement of both the new and old embankments. Furthermore, the MSE–FLS retaining wall widening method, as described in this study, outperforms the MSE widening scheme without FLS. The MSE–FLS retaining wall widening structure described in this research exhibits reduced settlement and differential settlement compared to the widening structure that just utilizes the MSE retaining wall. For instance, the application of FLS as a bedding layer resulted in a 22% reduction in vertical settlement at the shoulder of the new embankment compared to the model utilizing solely MSE retaining walls, while the maximum differential settlement decreased by 29%. In contrast to [9], the MSE–FLS retaining wall proposed in this paper exhibits reduced settlement, improved cooperative deformation between the old and new embankments, and a similar maximum differential settlement with Xie et al.’s [9] approach. Additionally, the proposed structure, with FLS as the bedding layer, will slope towards the shoulder after road operation which facilitates road drainage, but the road surface remains intact. Nevertheless, the expansion method put out [9] will result in minimal differential settlement at the intersection of the existing and new embankments. This, in turn, will prevent pavement cracking and other related issues, ultimately ensuring the safety of traffic.

### 3.2. Effect of Thickness of FLS

Figure 6A shows the vertical settlement of the road surface under different thicknesses of FLS. As shown in the figure, the higher the value of the thickness of FLS, the smaller the value of the settlement. This is because the thicker the FLS is, the lighter the new embankment will be, thus leading to a smaller vertical settlement. Meanwhile, for a given thickness of FLS, the settlement tends to be higher in new embankments. This is due to the fact that the void ratio in the new embankment is bigger than the ratio in an old embankment. Thus, the settlement tends to be larger under the same external loading say traffic loading. Figure 6B shows the vertical settlement of road surface under different bedding thicknesses of FLS. As shown in the figure, the higher the value of the thickness of FLS, the smaller the value of the settlement. This is because the thicker the FLS, the stronger the new embankment will be, and therefore the surface settlement of the embankment can be effectively reduced.

### 3.3. Effect of Density of FLS

According to the technical specification for cast-in-place foam lightweight soil, CECS 249:2008 [34], the density of foam lightweight soil is categorized into fourteen classes from D300 to D1600, as shown in Table 5. This subsection chose the density grade from D300 to D1200 to investigate the effect of the density of FLS on the performance of the MSE–FLS retaining wall.

Figure 7 illustrates the vertical settlement of the road surface under various densities of FLS fill, considering FLS as both the top layer and bedding layer. Figure 7 demonstrates that the vertical settlement of both the old and new embankments increases as the FLS density increases. Additionally, the maximum difference in the settlement also increases, regardless of whether FLS is utilized as the top layer or bedding layer. Figure 7A demonstrates that the settlement at the junction of the new and old embankments is discontinuous when FLS is utilized as the top layer fill. The reason for this is that when FLS is utilized as the uppermost layer, the self-weight is significantly lighter than the previous embankment fill. As a result, the deformation caused by the combined effect of self-weight and traffic load will be less than that of the old embankment. Therefore, the deformation is discontinuous at the conjunction of the old and new embankments. It is important to mention that when FLS is utilized as the bedding layer (as shown in Figure 7B), the deformation at the conjunction between the existing and new embankments is uninterrupted. The reason for this is that when FLS is used as a bedding layer fill, the new embankment’s overlying fill material forms a stronger connection with the junction of the old embankment. As a result, the old and new embankments deform in a way that complements each other. However, FLS is lighter than the old embankment fill material, which leads to a greater overall settlement in the new embankment compared to the old embankment. This phenomenon indicates that it is important to give priority to the design of FLS as a bedding layer during the construction of this structure in order to meet the settlement requirements at the intersection of the new and old embankments.

### 3.4. Effect of Leveling Pad of Facing Panel

Figure 8 depicts the vertical settlement of the road surface at various depths of burial for the leveling pad of the facing panel, taking into account the use of FLS as both the top layer and bedding layer. From the figure, it is evident that increasing the burial depth has minimal impact on the surface settlement of the old embankment, given a specific leveling pad size. However, the surface settlement of the new embankment will be reduced, resulting in a significantly smaller vertical settlement compared to the scenario without the pad. The pad’s setup will create an anti-tilting effect, enhancing protection against the panel’s overturning. The reason for placing the pad is to enhance the anti-tilting effect, which becomes more pronounced as the depth grows. Regardless of the depth of the pad, the unequal settlement of the old and new embankments may still be observed when FLS is utilized as the top layer. The utilization of FLS as a bedding layer enhances the synchronized deformation performance of both the existing and new embankments. When using this structure in real-world applications, the pad’s depth doesn’t need to be too deep to achieve enhanced performance.

Figure 9 depicts the vertical settlement of the road surface under various widths of the leveling pad of the facing panel, taking into account the use of FLS as both the top layer and bedding layer. From the figure, it is evident that increasing the width of the leveling pad leads to a decrease in road surface settlement, as well as a decrease in differential settlement between the old and new embankment. The reason behind this is that increasing the width of the pad reduces the pressure exerted on the soft clay layer, resulting in less deformation of the clay layer. However, if the width of the pad exceeds 1 m, increasing its width does not have a major impact on settlement control. This implies that purposely increasing the width of the pad is not required to control settlement during the construction of the structure. Observing Figure 9A, it is evident that uneven settlement is still present. However, in Figure 9B, a more favorable synergistic deformation between the old and new roadbeds is visible.

Figure 10 depicts the vertical settlement of the road surface under varying heights of the leveling pad of the facing panel, taking into account the use of FLS as both the top layer and bedding layer. By observing the figure, it is observed that increasing the height of the leveling pad results in a decrease in road surface settlement. Additionally, it leads to a reduction in differential settlement between the old and new embankments. The height of the pad directly affects its self-weight and the compression of the soft clay layer. A bigger self-weight improves the facing panel’s resistance to overturning. The resistance effect to overturning is more obvious, leading to reduced settling on the surface when the pad height is increased. However, if the height of the pad exceeds 1 m, increasing its height does not have a major impact on settlement control. This implies that increasing the height exceeding 1 m of the pad is not required in order to manage settlement during the construction and operation of the structure. Figure 10A demonstrates the continued presence of uneven settlement, whereas Figure 10B shows the improved synergistic deformation between the old and new embankments.

### 3.5. Effect of Modulus of the Soft Clay Layer

Figure 11 illustrates the vertical settlement of the road surface under a different modulus of soft clay foundation, considering FLS as both the top layer and bedding layer. The figure clearly demonstrates that the settlement of the road surface decreases as the modulus of the foundation increases, regardless of whether FLS is used as a top layer or a bedding layer. This is because a higher foundation stiffness leads to less deformation, and vice versa when subjected to the same overlying load on the foundation. Figure 11A illustrates the settlement curve of the road surface when FLS is utilized as the top layer. In contrast, Figure 11B displays the settlement curve of the road surface when FLS is utilized as the bedding layer. These lines have a similar shape, and the settlement curve of the surface with a high foundation modulus can be nearly aligned with the settlement curve of the surface with a low foundation modulus by shifting downwards. Furthermore, it is evident from Figure 11 that the displacement at the junction of the old and new embankments remains uninterrupted when FLS is employed as a bedding layer. This continuity can be attributed to the same rationale discussed in Section 3.2, which will not be reiterated here.

Figure 12 illustrates the horizontal of the facing panel under a different modulus of soft clay foundation, considering FLS as both the top layer and bedding layer. From the figure, it is evident that an increase in the modulus of the foundation results in a decrease in the horizontal displacement of the facing panel. This is due to the fact that a higher modulus of the foundation leads to a smaller overall deformation of the new and old embankments, resulting in a smaller horizontal displacement. Using FLS as the top layer (see Figure 12A) increases the total weight of the new embankment compared to using FLS as the bedding layer (see Figure 12B). Additionally, the horizontal earth pressure from the old embankment on the new embankment causes a smaller horizontal displacement of the facing panel. Nevertheless, the increased weight of the new embankment will lead to amplified subsidence of the foundation and increased displacement in the horizontal direction. When MSE–FLS retaining walls are utilized, it is important to focus on improving the horizontal stiffness of the walls. This is because the horizontal displacement is greater when FLS is used as the bedding layer, indicating the need for attention in this area.

## 4. Structure Design Using Modified HSA

Section 3 has demonstrated that the proposed MSE–FLS structure exhibits good performance in controlling differential settlement and horizontal displacement of the facing panel compared to the conventional approach. This section aims to provide a direct design of the proposed structure for practical purposes. It should be noted that MHSA will be presented in this section aiming to give an automatic design of the structure.

The specific steps are as follows:(1)Initialization: setting parameters, such as harmony memory size (HMS), harmony memory considering rate (HMCR), pitch adjusting rate (PAR), band width (BW), number of improvisation (NI), permutation evaluation rate (PER), etc.;(2)Initialization of harmony memory (HM): by determining the range of the vectors **S***_i_*′ and **L***_i_*′, which is contained in the HM, a solution space is determined. Then, HMS solutions are randomly generated in the solution space and stored in the HM;(3)Improvisation of a new harmony memory **H***_i_*′ = [**S***_i_*′, **L***_i_*′|*f*(**H***_i_*′)]: the vectors **S***_i_*′ and **L***_i_*′ in the new harmony are generated by four rules: memory considering, random choosing, pitch adjusting, permutation evaluation;(4)Estimate if there is any new harmony that is better than a stored harmony in HM. If yes, update the HM and estimate whether it meets the termination criteria. If not, repeat the step (3).

### 4.1. Structure Description

Before giving a brief introduction to the proposed algorithm, it is important to describe the retaining structure. As depicted in Figure 13, the structure comprises an unequal length of fill layer, with one or more layers filled with FLS instead of being filled with the same material, such as filler soil. Consequently, the established HSA requires modification to meet the requirement of calculating safety factors. The earth pressure of the old embankment on the newly filled embankment consists of two components: the first part is induced by overloading applied to the surface of the retaining wall, while the second part is active earth pressure acting on the newly filled embankment.

The MHSA follows the framework of traditional HSA, aiming to minimize costs while ensuring both global and local stability of the retaining wall. As shown in Figure 14 and Figure 15, global stability means that there is no sliding, overturning, or bearing capacity failure of the structure, while local stability ensures that there is no breaking or pulling out of the reinforcement. The primary difference between the MHSA and OHSA lies in the methods used to calculate these safety factors.

It is important to note that during the service of the MSE retaining wall, the soil may be removed through natural or man-made actions, such as various erosive actions, installation of utilities, etc. Consequently, passive earth pressures *P_a_* at the toe of the wall are disregarded in this section [35].

### 4.2. Design of MSE–FLS Retaining Walls

The MSE wall must be evaluated for both global and local stability. According to the Manual [35], three geotechnical failure modes should be taken into account simultaneously for global stability: overturning stability, sliding stability, and bearing capacity. In addition to global stability, local stability can be determined by calculating the pullout safety factor and the minimum layer spacing to meet the code [32]. For the detailed derivation procedure on the assurance of both global and local stability, please refer to the Appendix A section.

The safety factors for the design of MSE retaining walls are as follows: 2.0 for overturning, 1.5 for sliding, 2.0 for bearing, 2.0 for pullout, and 1.5 for reinforcement strength [32]. The design constraints in this section are considered to satisfy both the minimum values of the individual safety factors as well as the minimum design values specified in the code.

The design constraints (*g*_1_ to *g*_4_) associated with the above factors of safety are shown in the following equations:(1)$$g_{1}={\mathrm{FS}}_{\mathrm{design}(\mathrm{overturning})}-{\mathrm{FS}}_{\mathrm{overturning}}\leq 0$$
(2)$$g_{2}={\mathrm{FS}}_{\mathrm{design}(\mathrm{sliding})}-{\mathrm{FS}}_{\mathrm{sliding}}\leq 0$$
(3)$$g_{3}={\mathrm{FS}}_{\mathrm{design}(\mathrm{bearing})}-{\mathrm{FS}}_{\mathrm{bearing}}\leq 0$$
(4)$$g_{4}={\mathrm{FS}}_{\mathrm{design}(\mathrm{pullout})}-{\mathrm{FS}}_{\mathrm{pullout}}\leq 0$$

It is also necessary to consider the minimum values of the design parameters specified in the code [35], and these constraints (*g*_5_ to *g*_8_) are shown below:(5)$$g_{5}=s_{\min}-s\leq 0$$
(6)$$g_{6}=T_{(u)i}-60(\mathrm{kN}/m)\leq 0$$
(7)$$g_{7}=l_{\mathrm{emin}}-l_{e}\leq 0$$
(8)$$g_{8}=l_{n+1}-l_{n}<0,\;n=1,\;2,\;\ldots ,\;NoG-1$$
where *s*_min_ is minimum reinforcement spacing in the specification; *s* is actual reinforcement spacing; *T*_(*u*)*i*_ is the allowable tensile strength of reinforcement; *l*_emin_, is the minimum effective length of reinforcement in the specification [35].

The calculated safety factors of the MSE–FLS retaining wall are needed to meet the requirement of minimum factors recommended by [32]. The minimum safety factors are summarized and listed in Table 6. Once the requirement of safety factors is satisfied, the objective function or minimum cost needs to be constructed. The prices of each part of the retaining wall are collected from local prices in Table 6 and set as an example to illustrate the application of MHSA.

After giving the prices of each part of the retaining wall, the total cost can be expressed as follows:(9)$$f(x)=c_{0}\times \sum V_{FLS}+c_{1}+c_{2}\times \frac{\gamma_{b}}{g}\times \sum V_{fill}+\sum_{i=1}^{NoG}c_{3}\times l_{i}+c_{4}H+c_{5}H$$

Combining with restrictive conditions on safety factors, the complete expression of MHSA can be eventually obtained as follows:(10)$$\min f(x)\;s.t.\;g_{j}\leq 0;\;j=1,2,...,8$$

Adopting the same punitive mechanism [32], the modified objective function *Φ*(*x*) is shown as follows:(11)$$\phi (x)=f(x)\left(1+K\times C\right)$$
where *K* is a constant coefficient that serves to increase the value of the objective function to allow MSE retaining walls that experience large violation coefficients to receive significantly higher costs when calculated through the objective function *Φ*(*x*) and is generally assumed to be *K* = 10 for most engineering problems [32]; and *C* is a segmented function, defined as follows:(12)$$C=\sum_{j=1}^{m}C_{j}\leftarrow \left\{\begin{array}{ll} C_{j}=g_{j} & \mathrm{if}\;g_{j}>\;0\\ C_{j}=0 & \mathrm{if}\;g_{j}\leq \;0 \end{array}\right.$$

### 4.3. Results and Discussions

In the MHSA process, each iteration comprises 20 distinct harmonies, a number arbitrarily chosen. These harmonies represent 20 different prices, with the minimum identified as the optimal cost and the mean referred to as the average cost. To determine the optimal cost, the algorithm initially generates 20 harmonies with varied random values. In each iteration, the harmony with the highest price is substituted, followed by the subsequent iteration. Upon completion of the iterations, the smallest value is recognized as the optimal cost. The terminal condition is determined by assessing whether the absolute difference between the average cost and the optimal cost is less than 1000, and further iterations are necessary until the terminal condition is satisfied.

The cost of the retaining wall against the number of iterations is depicted in Figure 16. The figure reveals that the average cost is approaching the optimal cost with the increase in the number of iterations. The primary reason for this phenomenon is that the higher prices in each iteration are eliminated by replacing the design parameters, such as the length of reinforcement and the height of reinforcement, based on the MHSA. Only design parameters associated with relatively lower prices remain during each iteration and are reconsidered in the next iteration. After several iterations, the prices calculated from different design parameters exhibit close agreement with each other, at which point the iteration of the MHSA is terminated. As a result, the average cost equals the optimal cost. After several rounds of iteration, one set of design parameters is selected and displayed in Figure 17. A five-reinforcement design plan with detailed length and location of each reinforcement can be directly observed in the Figure 17. This program can serve as a reference for engineers and potential users.

## 5. Conclusions

In this study, the main conclusions in this paper are summarized as follows:A new MSE–FLS retaining wall structure was proposed for widening the existing road based on the review of the advantages and disadvantages of existing road widening methods;The established numerical model was compared with centrifuge tests. The results between the numerical model and experimental data show good agreement, which demonstrates the correctness and reliability of the numerical model;The computational results from the numerical model indicate that compared to conventional pile-supported embankments, utilizing FLS as a bedding or fill material for MSE retaining walls can decrease the average vertical settlement of the pavement by 9.3 cm and 9.8 cm, respectively. Utilizing FLS as a bedding for the MSE retaining wall can diminish the maximum differential settlement by 27.3%, decrease settlement at the interface of old and new embankments by 4%, and reduce settlement at the road shoulder by 20.1%. When FLS is utilized as a fill material, the relevant ratios are 27.4%, 36.1%, and 16.9%;The parametric research indicates that using FLS as a bedding layer in the proposed MSE–FLS construction improves road drainage and maintains the structural integrity of the road, hence representing a more beneficial setting. Moreover, when compared to a retaining wall absent of FLS incorporation, the application of FLS as a fill material with thicknesses of 1.5 m, 3 m, and 4.5 m leads to a decrease in maximum differential settlement by 28.4%, 53.5%, and 76.5%, respectively. The implementation of FLS as a bedding layer results in decreases of 26.4%, 55.4%, and 80.2%, respectively. Moreover, controlling the density grade of FLS at D300 can more efficiently alleviate the maximum differential settling at the new embankment;A modified HSA is proposed based on the newly developed retaining wall. The results show that the retaining wall structure can be quickly designed at the lowest cost by employing the modified HSA. The engineers and potential users can quickly design the retaining wall using the algorithm.

## Figures and Tables

**Figure 1 materials-17-05432-f001:**
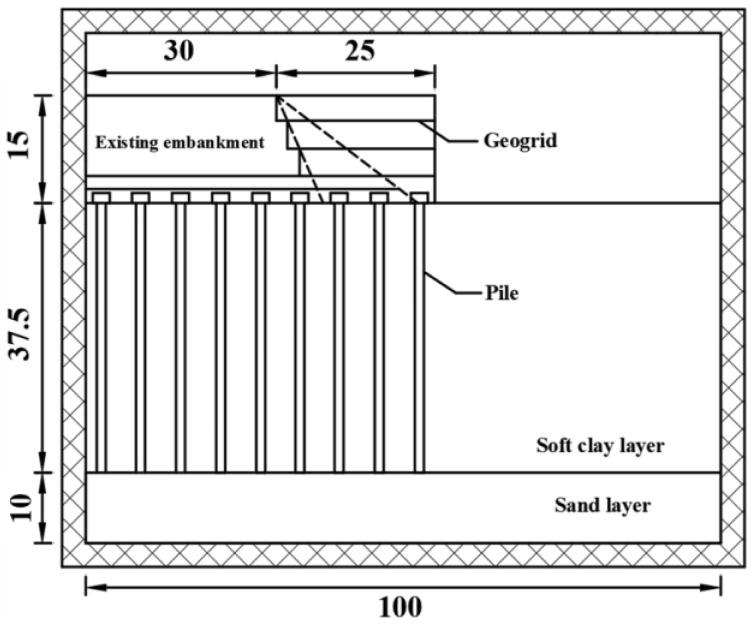
Schematic diagram of centrifuge model of the pile-bearing reinforced soil retaining wall (unit: cm).

**Figure 2 materials-17-05432-f002:**
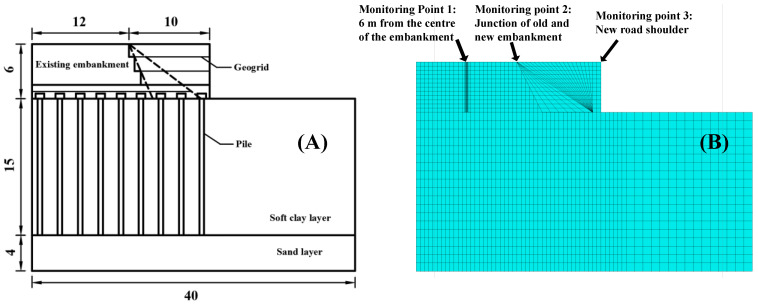
Schematic diagram of the model (unit: m): (**A**) numerical model; (**B**) calculation model.

**Figure 3 materials-17-05432-f003:**
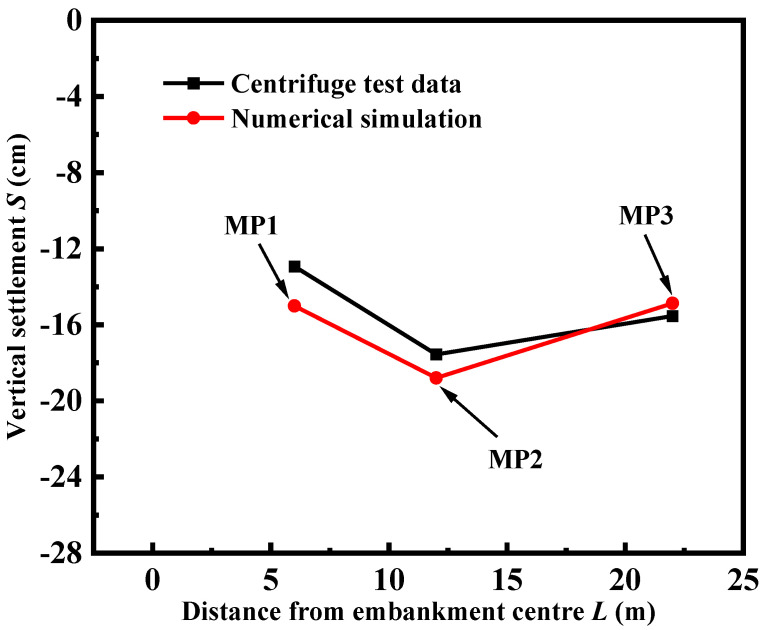
Comparison of centrifuge test data with numerical results.

**Figure 4 materials-17-05432-f004:**
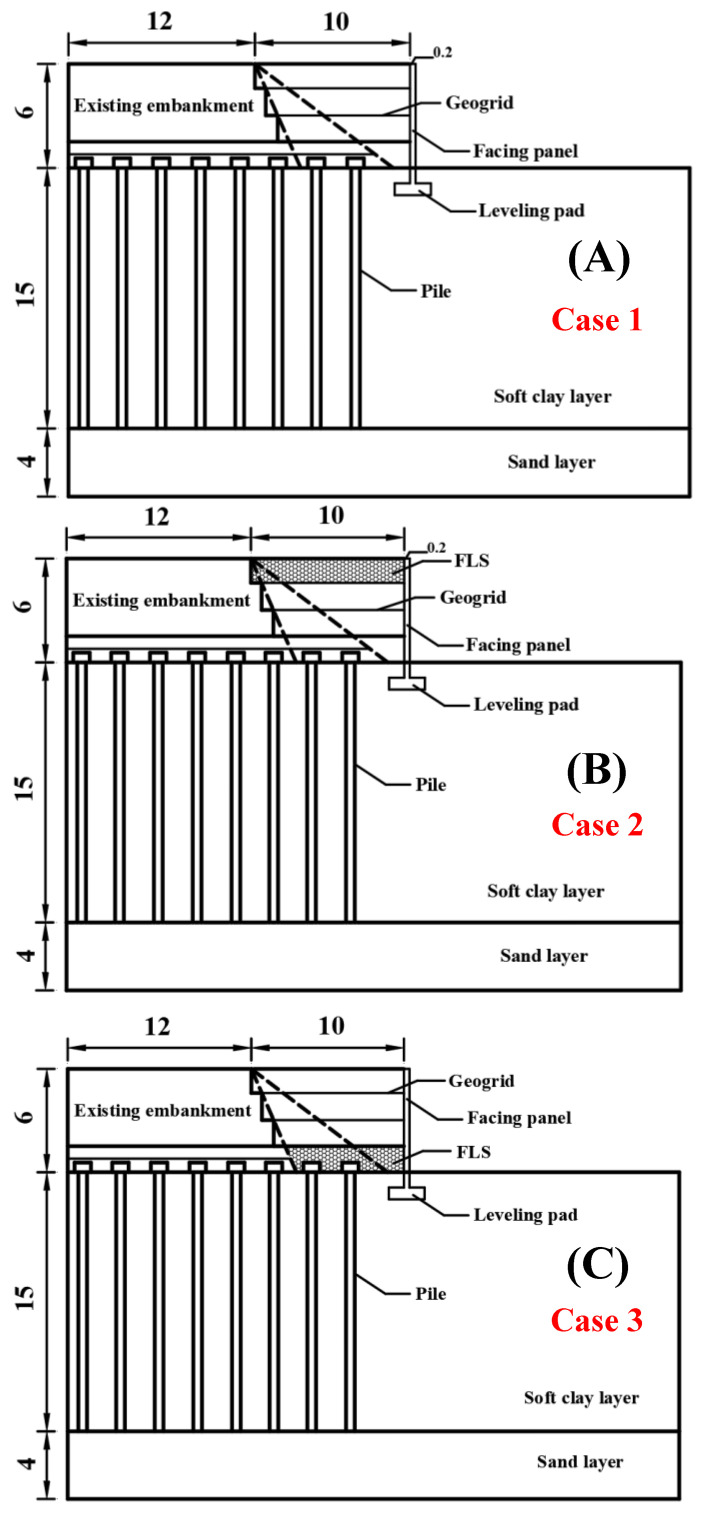
Schematic diagram of test model: (**A**) without FLS; (**B**) FLS as a subgrade fill; (**C**) FLS as a bedding layer.

**Figure 5 materials-17-05432-f005:**
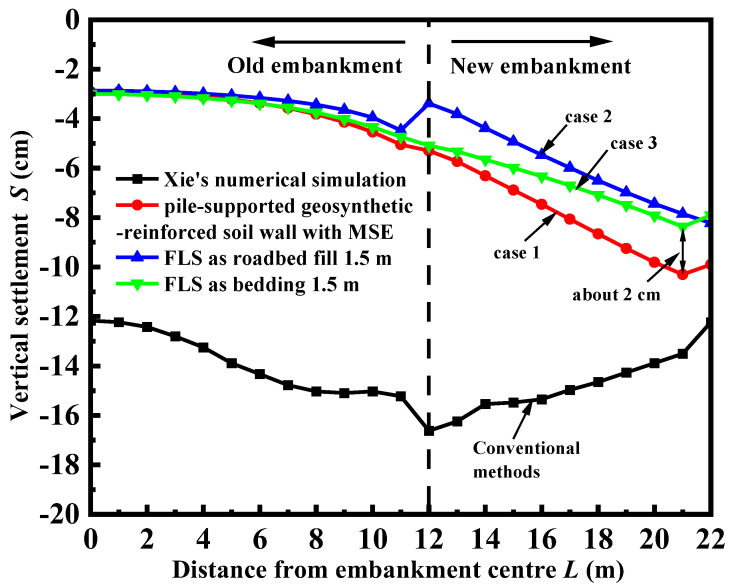
Vertical settlement of pile bearing reinforced soil retaining wall compared to case 1–3.

**Figure 6 materials-17-05432-f006:**
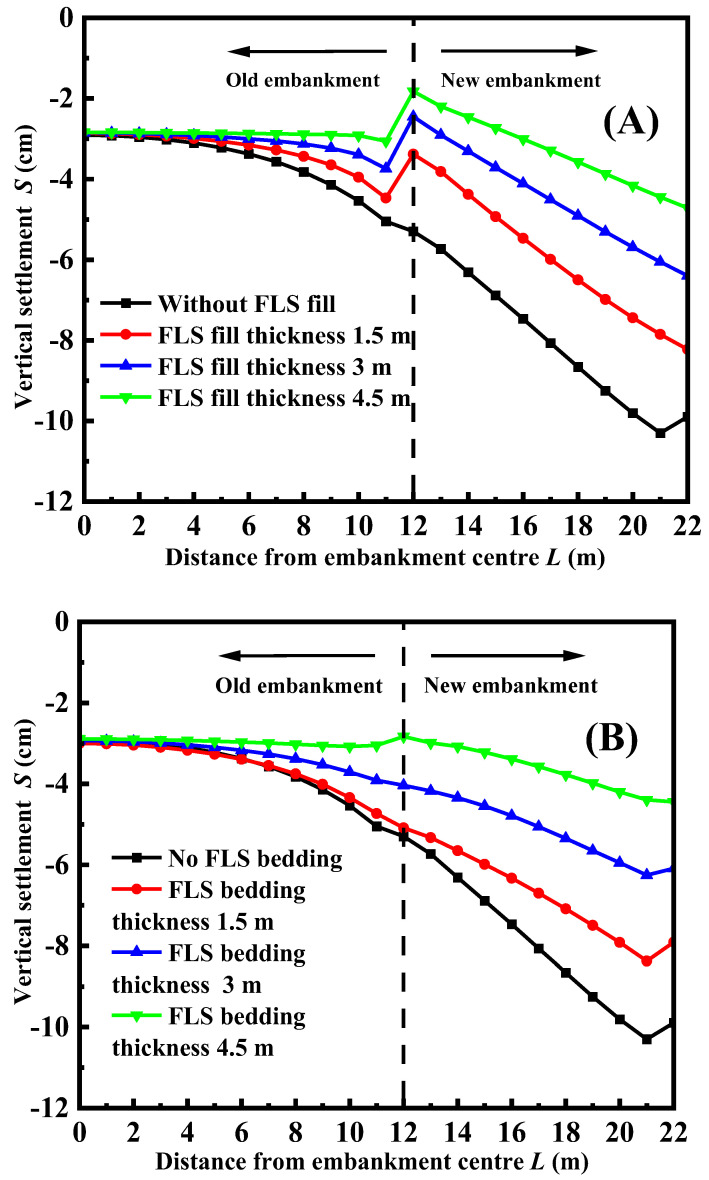
Vertical settlement of road surface with different thicknesses of FLS: (**A**) FLS is used as the fill; (**B**) FLS is used as a bedding layer.

**Figure 7 materials-17-05432-f007:**
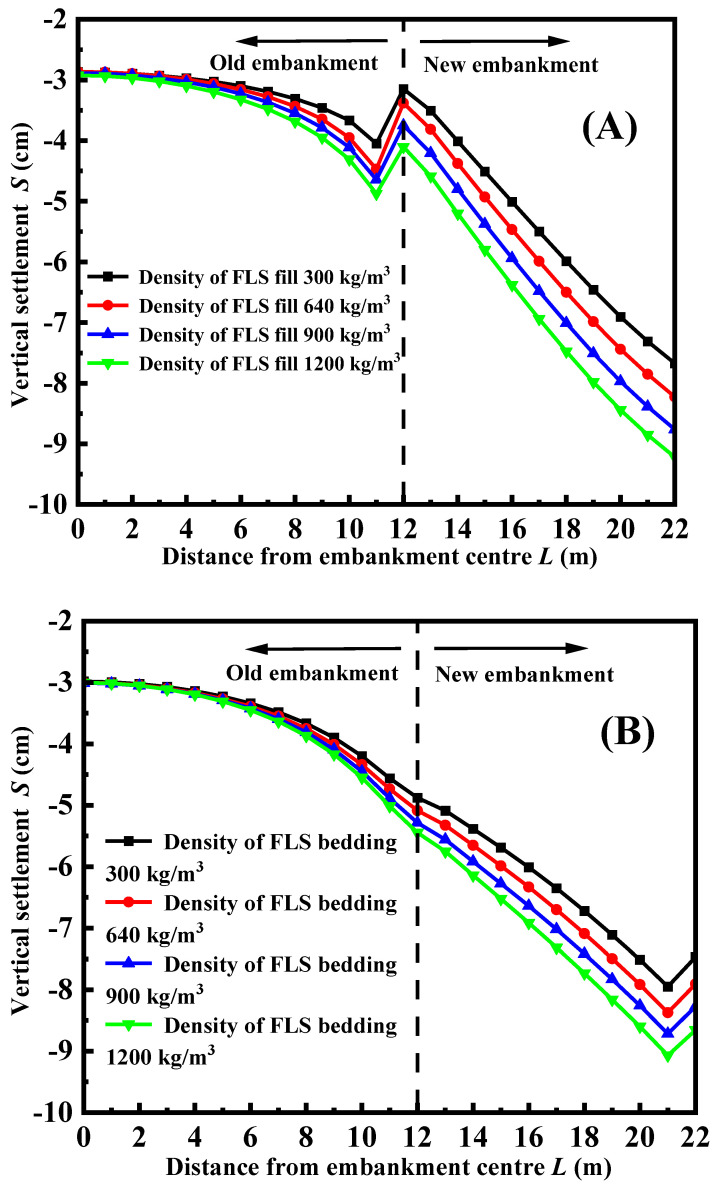
The vertical settlement of road surface with different densities of FLS: (**A**) FLS is used as the fill; (**B**) FLS is used as a bedding layer.

**Figure 8 materials-17-05432-f008:**
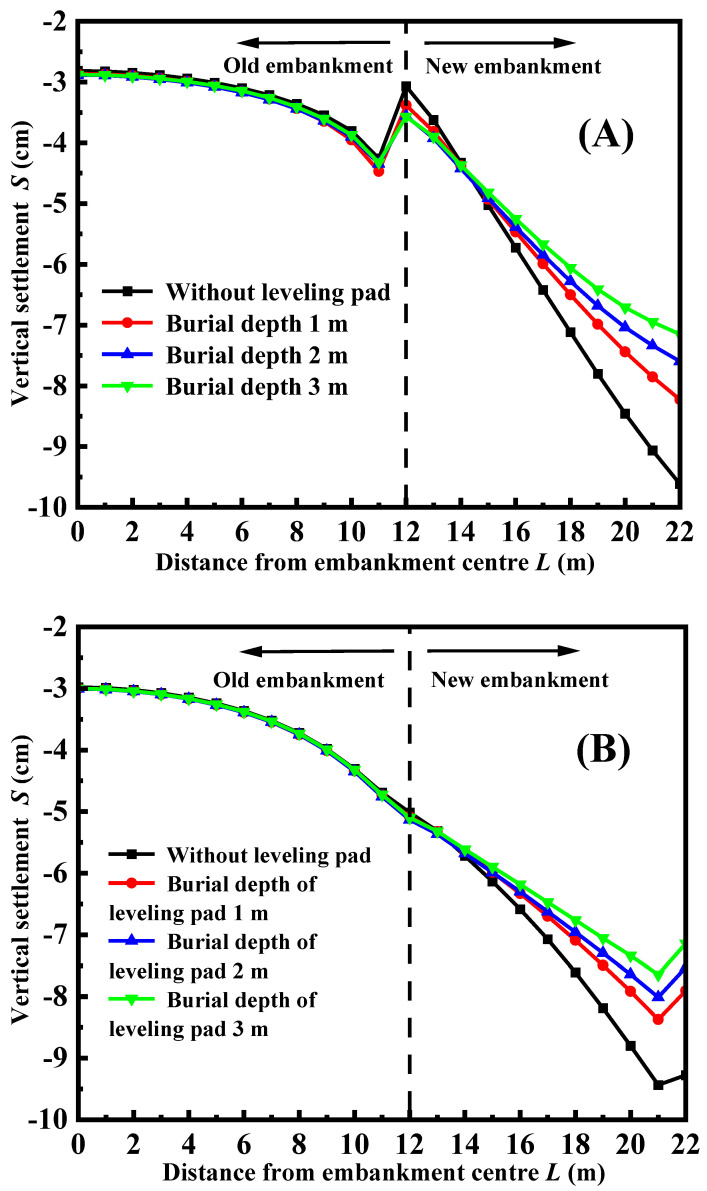
The vertical settlement of road surface with different foundation burial depths: (**A**) FLS is used as the fill; (**B**) FLS is used as a bedding layer.

**Figure 9 materials-17-05432-f009:**
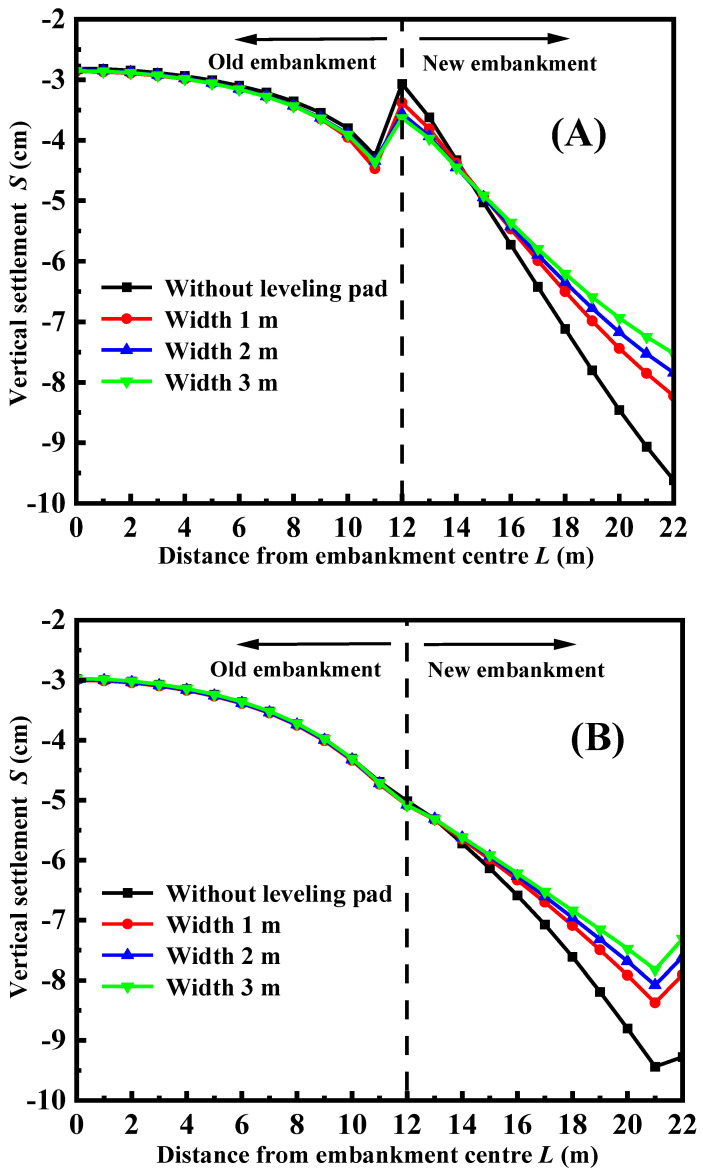
The vertical settlement of road surface with different widths: (**A**) FLS is used as the fill; (**B**) FLS is used as a bedding layer.

**Figure 10 materials-17-05432-f010:**
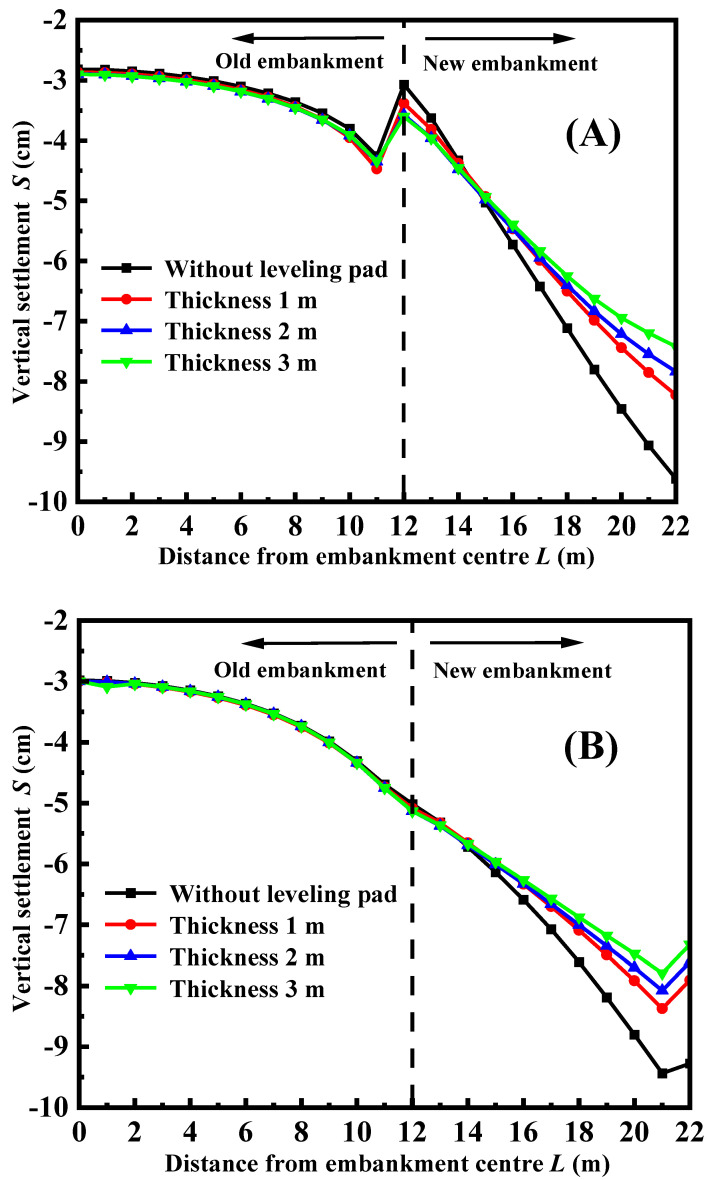
The vertical settlement of road surface with different foundation burial depths: (**A**) FLS is used as the fill (**B**) FLS is used as a bedding layer.

**Figure 11 materials-17-05432-f011:**
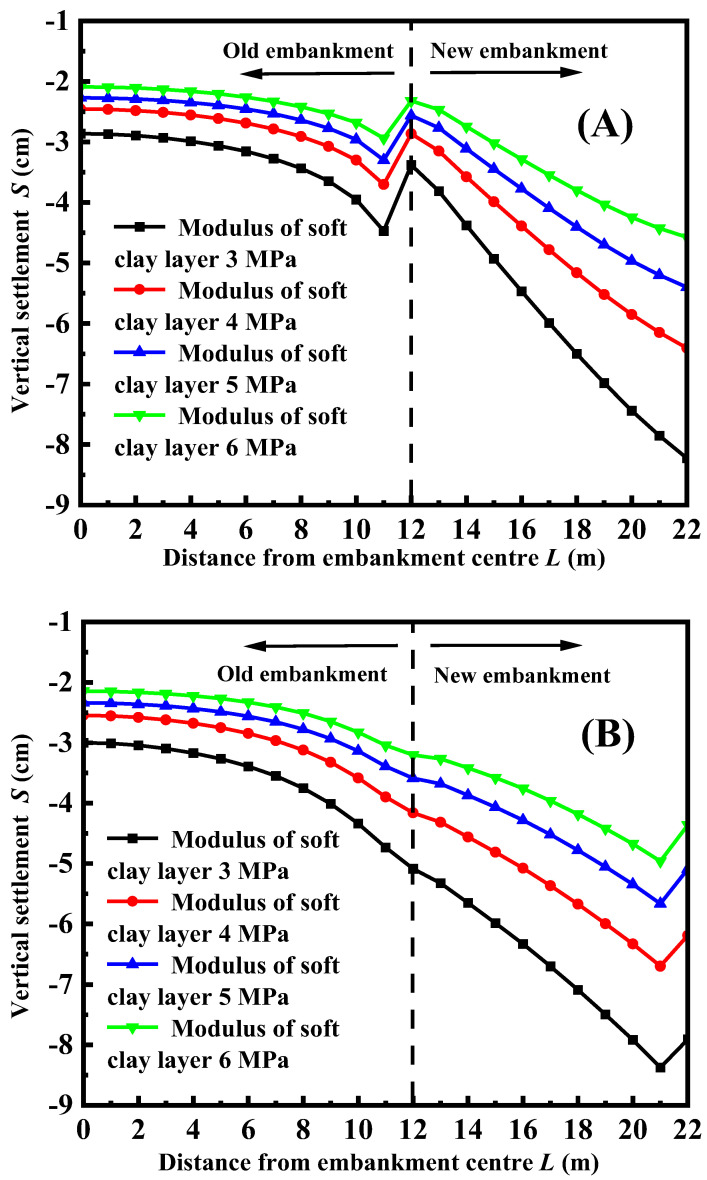
Variation of settlement curves of the top surface with different soft ground modulus: (**A**) FLS is used as a bedding layer; (**B**) FLS is used as the fill.

**Figure 12 materials-17-05432-f012:**
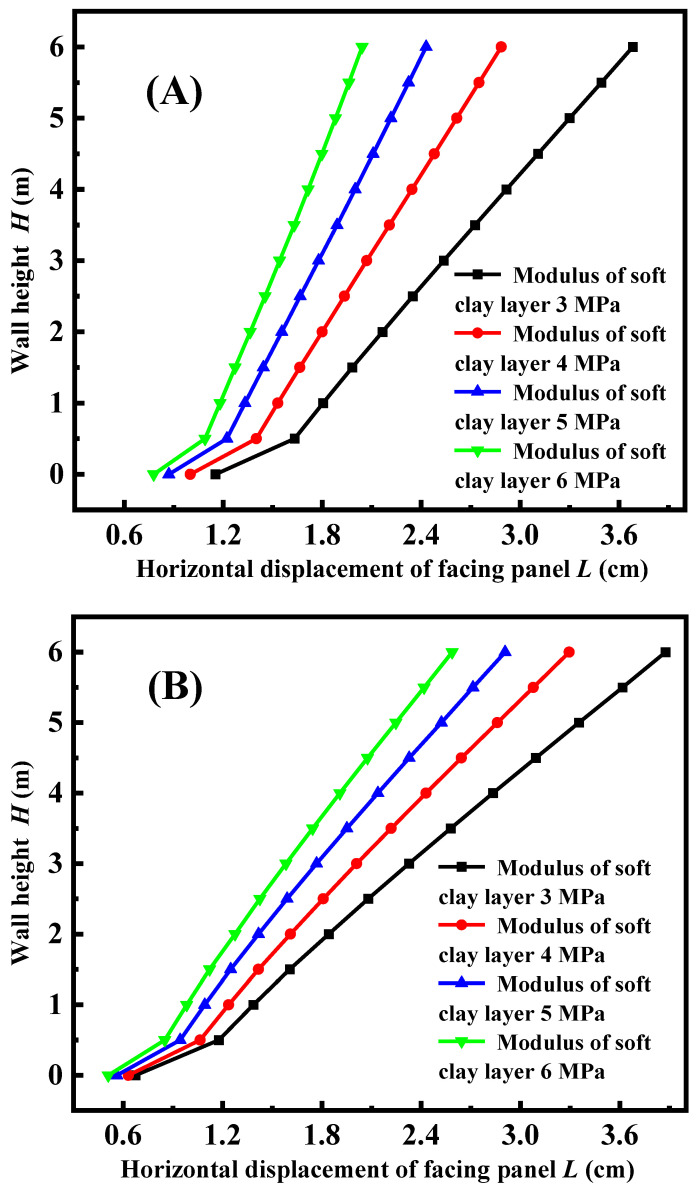
Variation of horizontal displacement curve with the different soft ground modulus: (**A**) FLS is used as a bedding layer; (**B**) FLS is used as the fill.

**Figure 13 materials-17-05432-f013:**
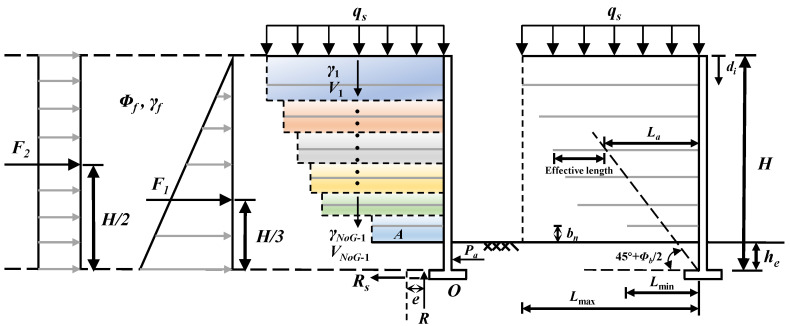
Schematic diagram of MSE–FLS retaining wall.

**Figure 14 materials-17-05432-f014:**
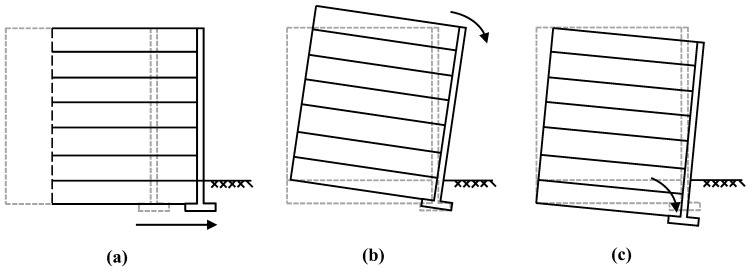
MSE retaining wall overall damage model: (**a**) sliding; (**b**) bearing capacity failure; (**c**) overturing.

**Figure 15 materials-17-05432-f015:**
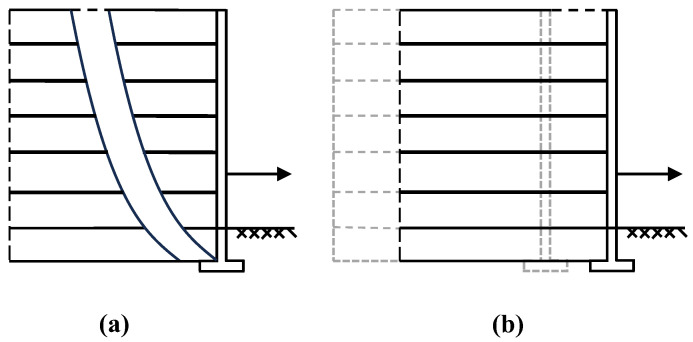
MSE retaining wall localized failure model: (**a**) break; (**b**) pullout.

**Figure 16 materials-17-05432-f016:**
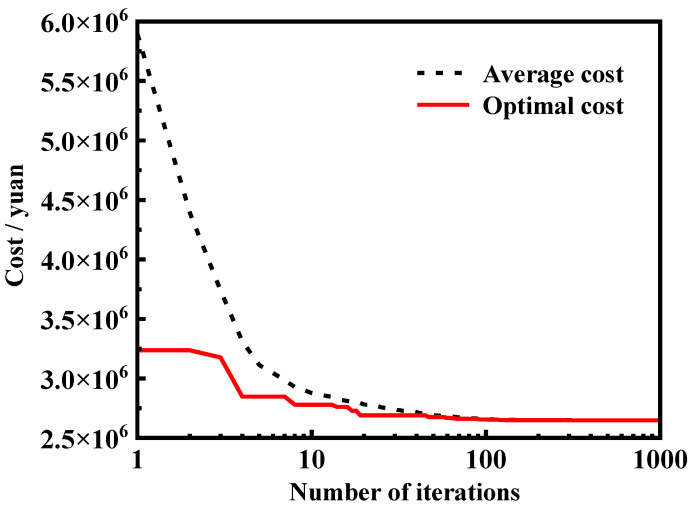
Harmony optimization results for the cost of the step MSE–FLS retaining wall figures.

**Figure 17 materials-17-05432-f017:**
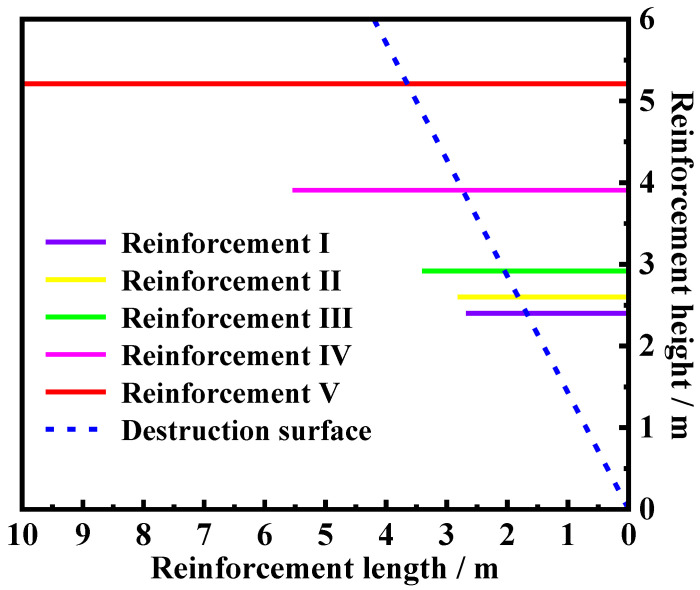
Arrangement of reinforcement materials for step-type MSE–FLS retaining walls.

**Table 1 materials-17-05432-t001:** Similarity scale for centrifugal test.

Classification	Physical Quantity	Symbol	Similarity Ratio
Geometric quantity	Length	*L*	1/*n*
Material properties	Heaviness	*γ*	*n*
	Cohesion	*c*	1
	Modulus of deformation	*E_S_*	1
	Flexural stiffness	*EI*	1/*n*^4^
	Compressive stiffness	*EA*	1/*n*^2^
External conditions	Velocity	*v*	1
	Acceleration	*a*	*n*
Properties	Displacement	*μ*	1/*n*

**Table 2 materials-17-05432-t002:** Parameters for the centrifuge model test of the soil.

Component	Parameters				
	Unit Weight, *γ* (kN/m^3^)	Elastic Modulus, *E* (MPa)	Poisson’s Ratio, *ν*	Friction Angle, *φ* (°)	Cohesion, *c* (kPa)
Embankment fill	19	6.3	0.35	16.6	7.5
Sand layer	16.5	50	0.32	30	0
Soft clay layer	18	3	0.42	25.5	59.5
Sand layer	16.5	50	0.32	30	0

**Table 3 materials-17-05432-t003:** Parameters for the centrifuge model test of the piles and geogrids.

Component	Parameters					
	Thickness, *d* (mm)	Elastic Modulus, *E* (MPa)	Poisson’s Ratio, *ν*	Friction Angle, *φ* (°)	Cohesion, *c* (kPa)	Densities, *ρ* (kg/m^3^)
Geogrid (clay layer)	5	50	0.3	11.5	30	1000
Geogrid (sand layer)	5	50	0.3	44.66	2.6	1000
Pile	0.5	40,000	0.2	-	-	2500
Pile cap	0.6	70,000	0.3	-	-	2700

**Table 4 materials-17-05432-t004:** Parameters for the parametric analysis.

Component	Parameters				
	Elastic Modulus, *E* (MPa)	Poisson’s Ratio, *ν*	Friction Angle, *φ* (°)	Cohesion, *c* (kPa)	Densities, *ρ* (kg/m^3^)
Foamed lightweight soil (FLS)	210	0.2	-	-	640
New roadbed fill	18	0.35	20	25	1800
C30 facing panel/Leveling pad	30,000	0.2	-	-	2350

**Table 5 materials-17-05432-t005:** Density of foam lightweight soil for different grades.

Density Grade	Range of Values of *ρ* (kg/m^3^)	Density Grade	Range of Values of *ρ* (kg/m^3^)
D300	250 < *ρ* ≤ 350	D1000	950 < *ρ* ≤ 1050
D400	350 < *ρ* ≤ 450	D1100	1050 < *ρ* ≤ 1150
D500	450 < *ρ* ≤ 550	D1200	1150 < *ρ* ≤ 1250
D600	550 < *ρ* ≤ 650	D1300	1250 < *ρ* ≤ 1350
D700	650 < *ρ* ≤ 750	D1400	1350 < *ρ* ≤ 1450
D800	750 < *ρ* ≤ 850	D1500	1450 < *ρ* ≤ 1550
D900	850 < *ρ* ≤ 950	D1600	1550 < *ρ* ≤ 1650

**Table 6 materials-17-05432-t006:** The prices associated with each part within the MSE–FLS retaining wall.

Component Type	Variable (Unit Price)	Total Cost per Item
FLS	*c*_0_ (200 yuan m^−3^)	$c_{0}\times \sum V_{FLS}$
Concrete foundation	*c*_1_ (430 yuan m^−1^)	*c* _1_
MSE backfill	*c*_2_ (60 yuan t^−1^)	$c_{2}\times \frac{\gamma_{b}}{g}\times \sum V_{fill}$
Geogrids	*c*_3_ ((2 + 0.03 *T_a_*) yuan m^2^)	$\sum_{i=1}^{NoG}c_{3}\times l_{i}$
C30 concrete panels	*c*_4_ (430 yuan m^−2^)	*c*_4_ × *H*
Labor cost	*c*_5_ (200 yuan m^−2^)	*c*_5_ × *H*

Noted: *T_a_* denotes the allowable tensile strength of the geogrid. Data source: https://b2b.baidu.com/ (accessed on 28 March 2024).

## Data Availability

The original contributions presented in the study are included in the article, further inquiries can be directed to the corresponding author.

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
