# Peer review of "Study on the Application of Foamed Lightweight Soil in Road Widening Project: A Numerical Insight"

_materials, 2024, doi:10.3390/ma17225432_

Round 1

Reviewer 1 Report

Comments and Suggestions for Authors

The reviewed manuscript titled "Study on the Application of Foamed Lightweight Soil in Road Widening Projects: Numerical Insights" is interesting, and the results presented are compelling. However, I would kindly suggest addressing the following minor points before considering it for acceptance in Materials:

1. Please provide more comprehensive comparisons between the analytical and experimental results, including a statistical correlation between the findings for clarity.

2. It is recommended to move equations (1) to (19), along with their explanations, to an appendix, as they are lengthy and not original contributions.

3. The methodology for estimating and comparing both the average and optimal costs is unclear. Kindly provide further clarification on this aspect.

4. The manuscript currently lacks sufficient references. Please consider adding more recent and relevant publications.

5. The conclusions section could be improved by providing more specific findings from the parametric study, as the current version reads more like a technical report.

Reviewer 2 Report

Comments and Suggestions for Authors

Dear Authors,

Thank you for your manuscript. The article discusses a novel retaining wall structure that combines a traditional Mechanically Stabilized Earth (MSE) wall with Foamed Lightweight Soil (FLS) as the fill material. The study evaluates the performance of this new structure through numerical modelling, verified by experimental data from a centrifuge test. The structure aims to reduce vertical and differential settlement between old and new embankments in road widening projects. A modified harmonic search algorithm (MHSA) is proposed for optimizing the design of this retaining wall, offering a cost-effective solution for engineers. The research concludes that this combined MSE-FLS structure improves stability and minimizes settlement, making it a practical approach for road expansions​. However, the paper cannot be published in the present form. In the following, some remarks are proposed that should be addressed to improve the quality of the paper. Therefore, a major revision is required before the paper can be accepted for publication.

Abstract

The abstract does not provide specific quantitative results or key findings, such as the degree of improvement in settlement reduction or cost savings from the new retaining wall structure

While it mentions the integration of MSE and FLS, it could be clearer on how this approach is novel compared to existing methods. 

Although the abstract refers to a numerical approach and the use of MHSA, it does not give the key insights into the specific methodologies used (e.g., the numerical models or centrifuge tests).

1. Introduction 

The introduction does not thoroughly explain the current challenges in road widening projects using traditional methods. 

Although the introduction references previous research, it lacks a clear identification of the specific research gap this study addresses.

The introduction touches on the novelty of combining MSE and FLS but does not provide enough detailed rationale for why this specific combination is innovative or superior to other methods.

The research objectives, while implied, are not stated explicitly or concisely in a way that frames the study.

2. Model establishment and verification

The section does not clearly elaborate on the assumptions made in the development of the numerical model. 

While the section provides some parameters, there is a lack of comprehensive details regarding all key input variables, such as exact soil properties, model dimensions, or any simplifying assumptions. 

There is no mention of a sensitivity analysis to assess how variations in key parameters affect the model’s output. 

While the section mentions validation with centrifuge tests, it lacks sufficient detail about the validation process. 

The section could benefit from more graphical representation of the comparison between the experimental results and the model. 

There is no discussion of potential uncertainties in the model, such as measurement errors in the experimental data or approximations in the numerical methods. 

The section does not mention any limitations of the model. 

3. Parametric analysis

The section does not sufficiently explain why the selected parameters (e.g., foundation width, FLS thickness) were chosen for analysis. 

There is a lack of analysis of the interaction between different parameters. The study only discusses how individual parameters affect performance, but many parameters likely interact in complex ways. 

While the section presents the effects of different parameters, it does not perform a sensitivity analysis to determine which parameters most significantly influence the performance of the structure. 

The parametric analysis seems to rely solely on numerical simulations, with no mention of how the results compare to real-world data or validated experiments

It would be helpful to compare the parametric results with other conventional road widening methods (e.g., pile-supported walls, geosynthetic reinforcement) to show how the proposed method performs relative to existing techniques. 

4. Structure design using modified HAS

The section does not provide enough clarity on how the modified harmonic search algorithm (HSA) differs from the traditional HSA. 

There is limited discussion of the specific steps involved in the algorithm’s process. 

The section does not compare the modified HSA with other optimization techniques. Including a comparative analysis with alternative optimization methods, such as genetic algorithms or simulated annealing, would demonstrate the efficiency and advantages of the modified HSA in this context.

The design constraints considered in the modified HSA are mentioned but not discussed in depth. 

There is no discussion of how well the modified HSA converges to an optimal solution.

The section includes a cost optimization discussion, but it does not explain how the costs were derived or what assumptions were made regarding material, labour, and construction costs. 

The section does not discuss any limitations of the modified HSA. For example, the algorithm may have limitations regarding scalability, computational complexity, or its application to more complex structures. 

It would be valuable to include sensitivity analysis to evaluate how robust the optimized design is to variations in key parameters. This would provide insights into how small changes in input parameters (e.g., material properties, loads) could affect the optimized design and the performance of the structure.

5. Conclusions

The conclusions are relatively general and do not provide specific quantitative results from the study

The conclusions do not provide suggestions for future research. The section focus heavily on the success of the modified harmonic search algorithm (MHSA) but could discuss more about the structure’s performance itself. 

Summarizing how the new approach outperforms or complements traditional retaining wall structures would give a stronger sense of the study’s contribution to the field.

Comments on the Quality of English Language

 There are several grammatical issues throughout the paper. For instance, there are incomplete sentence constructions, inconsistent tense usage, and missing articles that impact readability.

Round 2

Reviewer 2 Report

Comments and Suggestions for Authors

Accept

Comments on the Quality of English Language

Accept